# Tissue culture and *Agrobacterium*-mediated genetic transformation of the oil crop sunflower

Fangyuan Chen[1,2], Youling Zeng[1]*, Quan Cheng[1], Lvting Xiao[1], Jieyun Ji[1], Xianfei Hou[3], Qixiu Huang[3], Zhonghua Lei[3]*

1 Xinjiang Key Laboratory of Biological Resources and Genetic Engineering, College of Life Science and Technology, Xinjiang University, Urumqi, XinJiang, China, 2 Key Laboratory of Microbial Resources Protection, Development and Utilization, College of Biological Sciences and Technology, Yili Normal University, Yining, XinJiang, China, 3 Institute of Economic Crops, Xinjiang Academy of Agricultural Sciences, Urumqi, XinJiang, China

* zeng_ylxju@126.com (YZ); youkui@sina.cn (ZL)

**Data Availability Statement:** All relevant data are within the paper and its Supporting information files.

**Funding:** This work was financially supported by the Natural Science Foundation of Xinjiang Uygur

## Abstract

Sunflower is one of the four major oil crops in the world. 'Zaoaidatou' (ZADT), the main variety of oil sunflower in the northwest of China, has a short growth cycle, high yield, and high resistance to abiotic stress. However, the ability to tolerate adervesity is limited. Therefore, in this study, we used the retention line of backbone parent ZADT as material to establish its tissue culture and genetic transformation system for new variety cultivating to enhance resistance and yields by molecular breeding. The combination of 0.05 mg/L IAA and 2 mg/L KT in MS was more suitable for direct induction of adventitious buds with cotyledon nodes and the addition of 0.9 mg/L IBA to MS was for adventitious rooting. On this basis, an efficient *Agrobacterium tumefaciens*-mediated genetic transformation system for ZADT was developed by the screening of kanamycin and optimization of transformation conditions. The rate of positive seedlings reached 8.0%, as determined by polymerase chain reaction (PCR), under the condition of 45 mg/L kanamycin, bacterial density of $OD_{600}$ 0.8, infection time of 30 min, and co-cultivation of three days. These efficient regeneration and genetic transformation platforms are very useful for accelerating the molecular breeding process on sunflower.

## Introduction

Sunflower (*Helianthus annuus* L.) is an important oil crops worldwide. Approximately 52 million tons of sunflower was cultivated, covering 27 million hectares in 2018; almost all of it was used for oil extraction and accounted for 9.0% of the total edible oil produced in the world [1]. Sunflower is also known as a 'pioneer crop for saline lands'. The complete genome sequence of sunflower was public in 2017 [2], and so tissue culture and genetic transformation for sunflower are very necessary for promoting the molecular biological study with these genetic information, including the improvement of stress tolerance and yields.

Autonomous Region (2020D01C020), Project 4 of Xinjiang Autonomous Region Major Special Project (2022A03004-4) and the Research and Innovation Program for Excellent Doctoral Students at Xinjiang University (XJU2022BS050). The funders had no role in study design, data collection and analysis, decision to publish, or preparation of the manuscript.

**Competing interests:** The authors have declared that no competing interests exist.

A well-established plant regeneration protocol is often a prerequisite for genetic transformation. Usually, genotype specificity influences the regeneration of most plants, such as cotton [3], sunflower [4], etc. Of the four *indica* rice varieties by mature seeds inducing callus and embryogenic cell, only one variety (ASD16) produced regenerated plants [5]. Regeneration conditions are determined by optimizing various factors, such as different explants, combination and concentration of plant hormones. Mullein (*Casuarina equisetifolia*) stem segments [6] and *garland chrysanthemum* leaf disks as explants [7] can induce callus, adventitious buds, and adventitious roots with different hormone combinations. Two regeneration pathways in sunflower were reported in the late 20th century, including organogenesis (immature embryos [8, 9], cotyledons [10–12], the thin cell layer of the hypocotyl [13], stem tips [14], dicotyledons [15], protoplasts [9]) and somatic cell embryogenesis (embryogenesis immature embryos [16, 17] and pollen sacs [18]), but these process were unsystematic and superficial. And then, regenerated plants were obtained by cotyledons of sunflower RHA 6D-1 seedlings as explants with 1.0 mg/L isopentenyl adenine (2-iP)+0.5 mg/L indoleacetic acid (IAA)+0.1 mg/L thidiazuron (TDZ), the regeneration efficiency was 45.8~71.4%, while many malformed buds were produced [19]. Therefore, establishing an effective regeneration system is very necessary for different genotype sunflower to research and application.

Genetic transformation is usually carried by *Agrobacterium*-mediated method, its transformation efficiency is often influenced by different factors, such as *Agrobacterium* strains, bacterial density, duration of infection, co-culture time, acetosyringone concentration, and the type of antibiotics [20], and so on. The gene was successfully transformed into *Hevea brasiliensis* using cotyledonary somatic embryos as explants [21], similarly, immature leaf segments of Switchgrass (*Panicum virgatum*) [22] and leaves of dandelion (*Taraxacum officinale*) were successfully transformed to obtain positive seedlings by optimizing various conditions [23]. Genetic transformation in sunflower is less reported. A research group used the *Agrobacterium tumefaciens* strain *GV2260* to infect shoot apices of sunflower inbred line HA300B by cell wall enzyme digestion and ultrasonic treatment for increasing the amount of *Agrobacterium* invading the organism. However, only two preliminary transformed seedlings were obtained [24]. And in another study, the split cotyledons from dehulled mature seeds was infiltrated by *Agrobacterium* to obtain approximately 3.3% of the putative transformed buds [19]. In a word, an efficient transformation platform needs to be established for sunflower.

In this study, the retention line of backbone parental ZADT was used as material to establish a good regeneration and genetic transformation system. Cotyledonary nodes as explants were used to induce adventitious buds with a combination of IAA and kinetin (KT), and *Agrobacterium tumefaciens* strain *EHA105* containing pBI121 with a *GUS* reporter gene and a selectable marker gene (*NPTII*) to establish transgenic system for ZADT. These platforms are efficient to study biological functions and the mechanisms of candidate genes in sunflowers and, to accelerate the molecular breeding process by biotechnology.

## Results

### Tissue culture and regeneration

**Induction of adventitious buds for 'ZADT'.** Four explants (hypocotyl, cotyledon, true leaves, cotyledon node) (Fig 1A–1D) were induced for adventitious buds on MS medium (S1 and S2 Tables) with 60 different combinations of plant growth regulators (6-BA, KT, and IAA). The results showed that a few hypocotyls and true leaves could be induced successfully, but the adventitious buds turned yellow or brown (Fig 1E and 1G). Among all the combinations, cotyledons had a highest induction rate (50%) at such a combination of 0.05 mg/L IAA and 5 mg/L 6-BA and normal adventitious bud phenotypes were shown in Fig 1F.

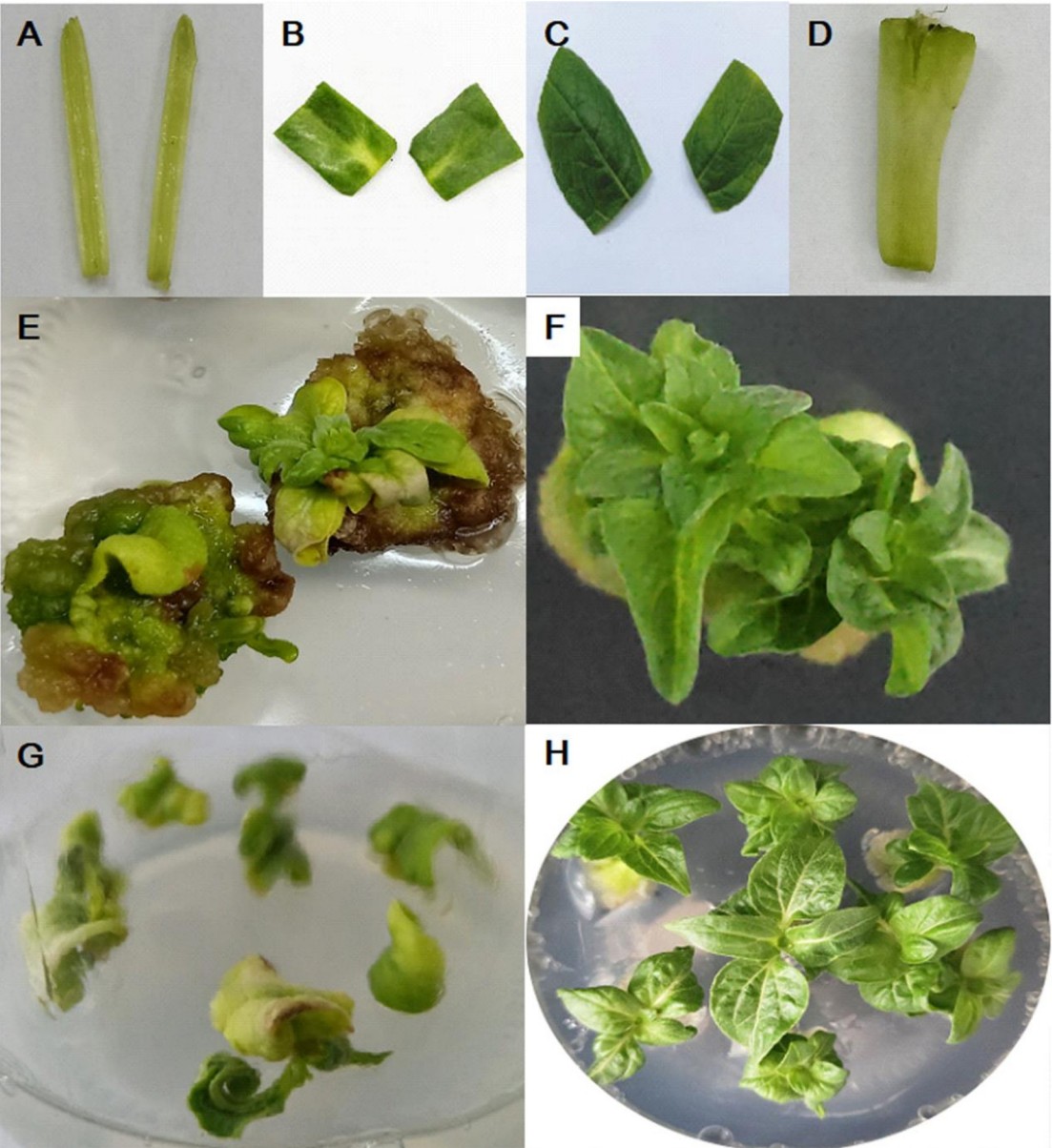

**Fig 1. Direct inducting adventitious buds by different explants from ZADT.** (A) Hypocotyl; (B) cotyledons; (C) true leaves; (D) cotyledon nodes; (E-H) the phenotypes of adventitious buds by the direct induction with hypocotyl, cotyledon, true leaves, cotyledon nodes.

However, compared to the above explants, the cotyledonary nodes were the best for the induction of adventitious buds (Fig 2A and 2B; S2 Table). The hormone combinations 1, 2, 11, 12, and 13 having high induced rates with 88.7%, 77.8%, 88.9%, 83.3%, and 94.4% for this process were renamed as B1 (IAA 0.05 mg/L, 6-BA 0.25 mg/L), B2 (IAA 0.05 mg/L, 6-BA 0.3 mg/L), B3 (IAA 0.05 mg/L, KT 0.5 mg/L), B4 (IAA 0.05 mg/L, KT 1 mg/L), and B5 (IAA 0.05 mg/L, KT 2 mg/L). But adventitious buds were not easy to root due to the curled leaves and the small seedlings in B1-B4. B5 is the most favorable for direct induction and rooting (Fig 1H).

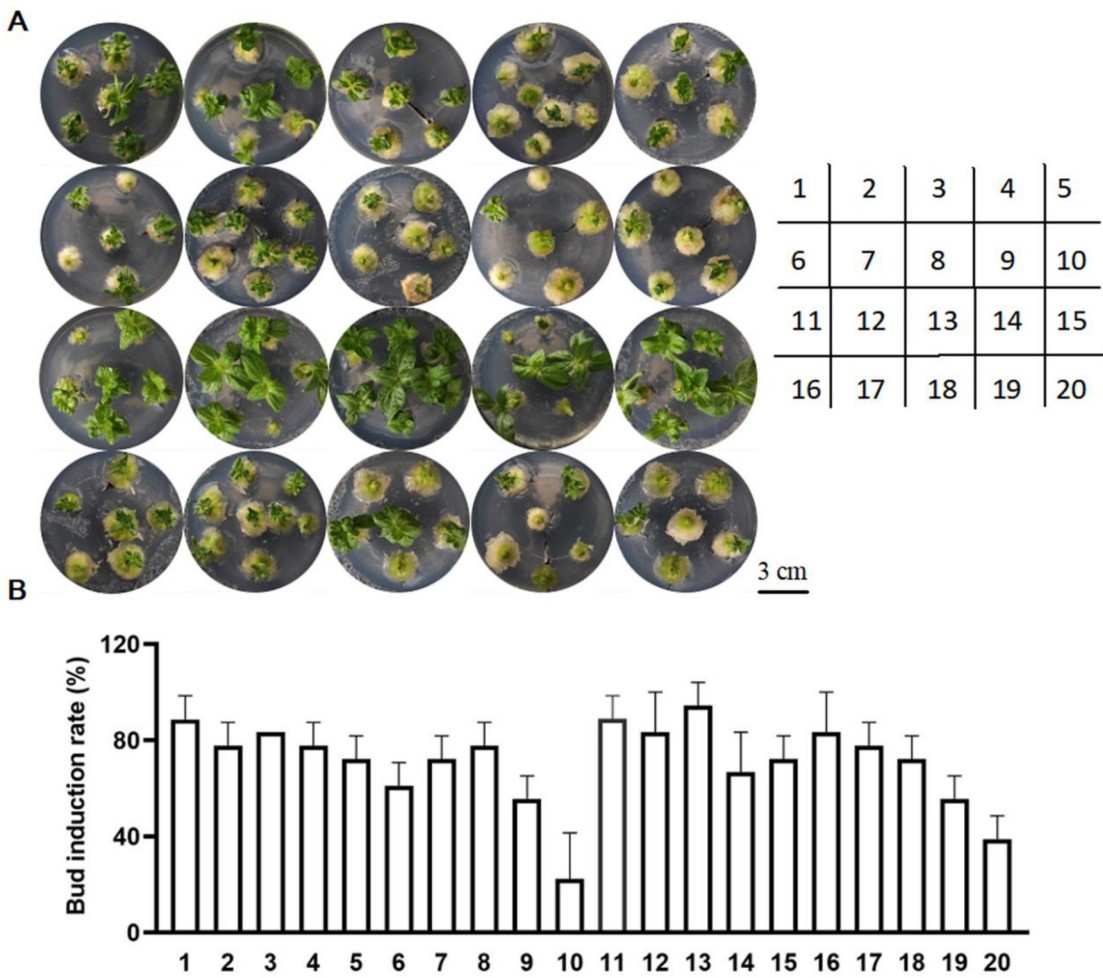

**Fig 2. Direct induction of adventitious buds from cotyledon nodes of ZADT.** (A-B) Phenotype and induction rate of adventitious buds by (plant growth regulator) PGR combinations; the datas were represented as the means±SE of three biological replicates.

**Optimization of adventitious bud subculture and rooting.** Adventitious buds needed to be subcultured to obtain 1 cm in length at least for subsequent rooting. Three media of no hormone (CK), 1/2 B5, and B5 were designed to compared the status of bud growth. The results showed that buds turned yellow to a lesser extent and more vigorous in 1/2 B5 than the other two media, and the B5 medium was the worst to make buds yellow to a significantly greater extent (Fig 3A). For rooting, subcultured buds were cultured on the MS medium containing different concentrations of Indole-3-butyric Acid (IBA). The R3 medium with 0.9 mg/L IBA was the best to induce 60% rooting and was initiated on the third day of transfer (Fig 3B–3D). R1 medium no adding IBA was the worst (33.3%) with a few roots per bud and rooting was induced on the seventh day (Fig 3B and 3C). To summarize, the regeneration of ZADT with cotyledonary nodes as explants to successfully be established in approximate 1.5 months (Fig 4).

## Genetic transformation of the sunflower ZADT with cotyledonary nodes

**Optimization for the transformation of cotyledonary nodes by *Agrobacterium tumefaciens*.** In this study, *Agrobacterium tumefaciens EHA105* and pBI121 vector with the *GUS*

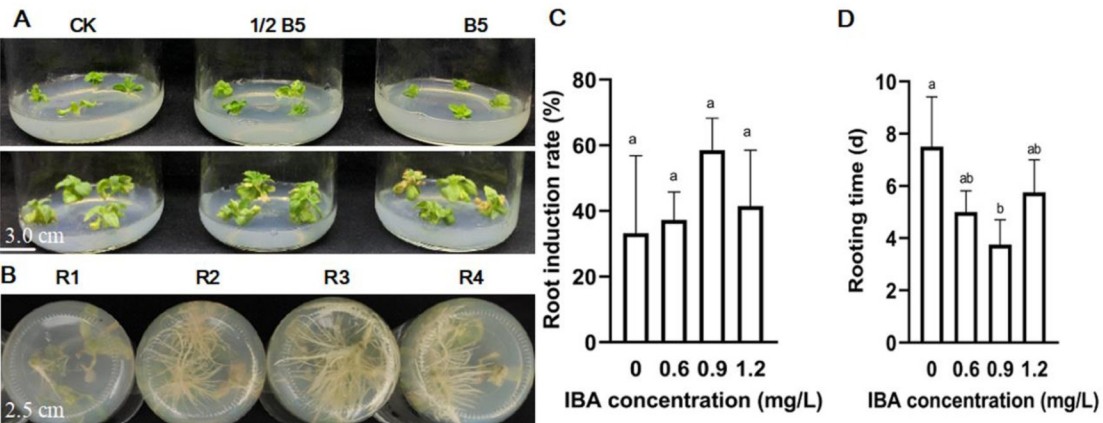

**Fig 3. The subculture and rooting of adventitious buds.** (A) Subculture culture of adventitious buds in MS with different combinations of PGRs. CK: no hormone, 1/2 B5: IAA 0.025 mg/L KT 1 mg/L, B5: IAA 0.05 mg/L KT 2 mg/L; (B) The rooting of adventitious buds in MS with different IBA concentrations (0.3 mg/L, 0.6 mg/L, 0.9 mg/L, and 1.2 mg/L), respectively; (C) and (D) induction rate and rooting time of adventitious roots; the data were represented as the means±SE of three biological replicates. The values on the columns followed by the same letter were not significantly different at $p < 0.05$ (Tukey test).

reporter gene and the *NPTII* gene were used for transformation. 45 mg/L kanamycin was well-determined for the screening of resistant buds by direct induction of the cotyledonary node (Figs 5A and 7C).

The transformation efficiency were further evaluated by three factors, including *Agrobacterium* density, infection time, and co-culture time. $OD_{600}$ 1.0 made the browning of resistant buds very serious, up to 48.9%. However, under the conditon of $OD_{600}$ 0.8, the resistant bud rate could also achieve the same as high with 25%, but the browning rate was only 29.5%, so this density was more suitable for transformation (Fig 5B). Secondly, for the determination of the infection time to reduce browing and increase more resistant buds, infection time of 30 min was optimal with the highest rate of resistant buds (28%) and the browning (37%), compared with the other those of 10 min and 20 min (Fig 5C). Finally, optimization of co-culture times (1~7 days) revealed that co-culture of 3 days was impotant for the highest rate of resistant buds with 26% (Fig 5D). Altogether, these conditions of $OD_{600}$ 0.8, infection time of 30 min, and co-culture of three days were confirmed for genentic transformation of ZADT with cotyledonary nodes by *Agrobacterium* mediated method.

**Identification of transgenic plants by β-glucuronidase (GUS) staining and PCR.** In order to detect transformation efficiency of ZADT, leaves and buds of resistant seedlings grown for four weeks under kanamycin-screening was underwent the GUS staining and PCR. The results showed that the strong GUS staining signal was presented in the resistant buds, but not in the control (Fig 6A–6F). Correspondingly, preliminary integrated statistics of the *GUS* gene by genomic PCR analysis suggested the positive rate was up to 8% from 475 infected cotyledonary nodes (Fig 6G).

In summary, the process of *Agrobacterium*-mediated genetic transformation for ZADT was as follows: cotyledonary nodes were infected for 30 min with a suspension of *EHA105* and cultured in a medium (MS+0.05 mg/L IAA+2 mg/L KT+150 μmol/L Acetosyringone (As)) for 3 d in the dark. Next, cotyledonary nodes were successively transferred into the resistance screening medium (MS+500 mg/L Cef+45 mg/L Kana+0.05 mg/L IAA+2 mg/L KT), subculture medium (MS+100 mg/L Cef+20 mg/L Kana+0.025 mg/L IAA+1 mg/L KT), and rooting medium (MS+100 mg/L Cef+0.9 mg/L IBA) for two weeks each (Fig 7). Finally, the plantlets

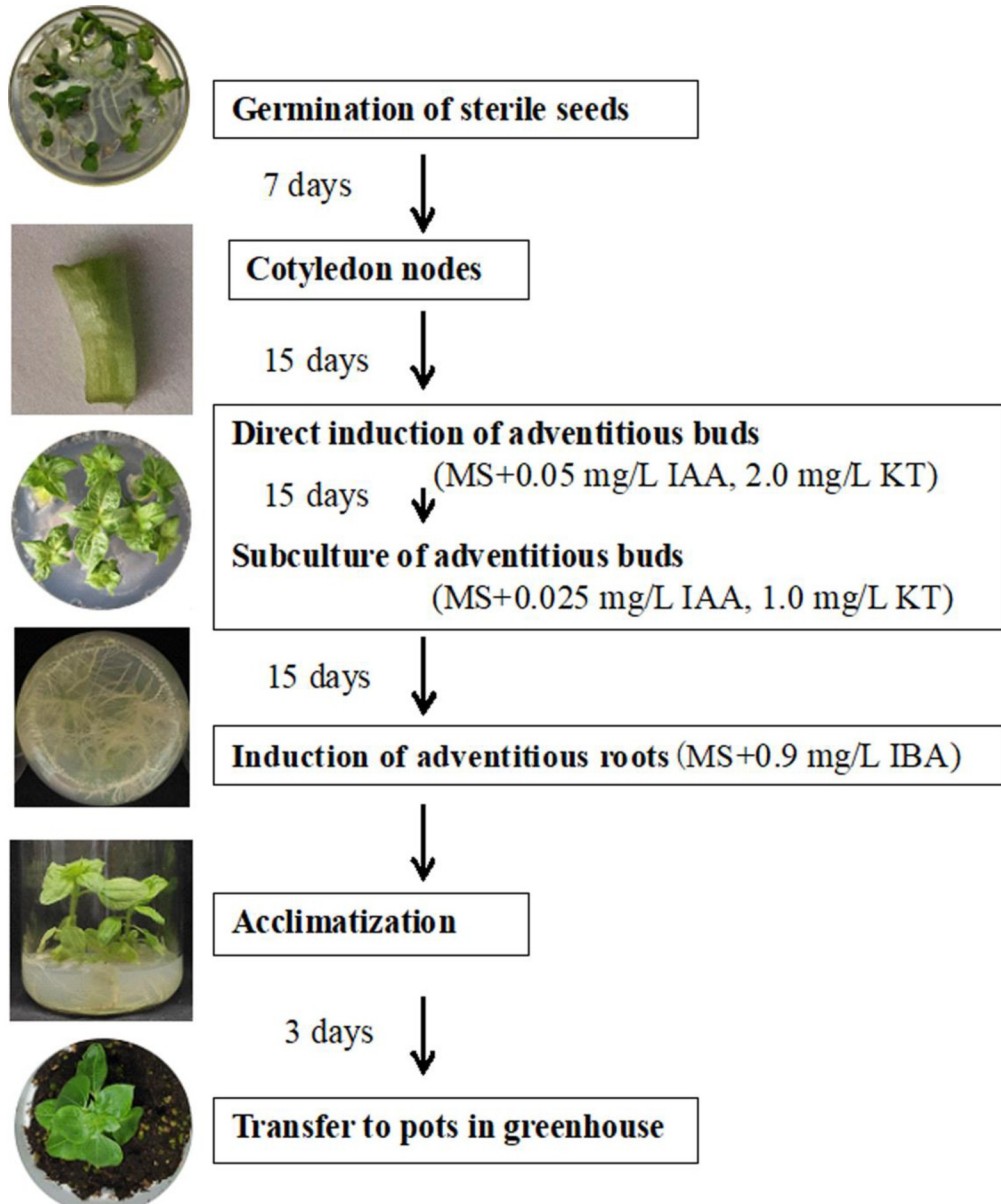

**Fig 4. Flow diagram of the tissue culture and regeneration for the ZADT.**

were domesticated for three days and transferred into pots and incubated under well-illuminated conditions (2,000 lx) at 28°C/16 h of light and 26°C/8 h of darkness. 18 of the 21 plants grown for 2 months were confirmed positive again through a new round of *GUS* gene testing at the genomic level. This suggested that transformation system for the genotype ZADT was successfully established.

## Discussion

The tissue culture and genetic transformation are quite necessary for different genotype sunflower. In this study, we used the retention line of oil sunflower backbone parent ZADT to

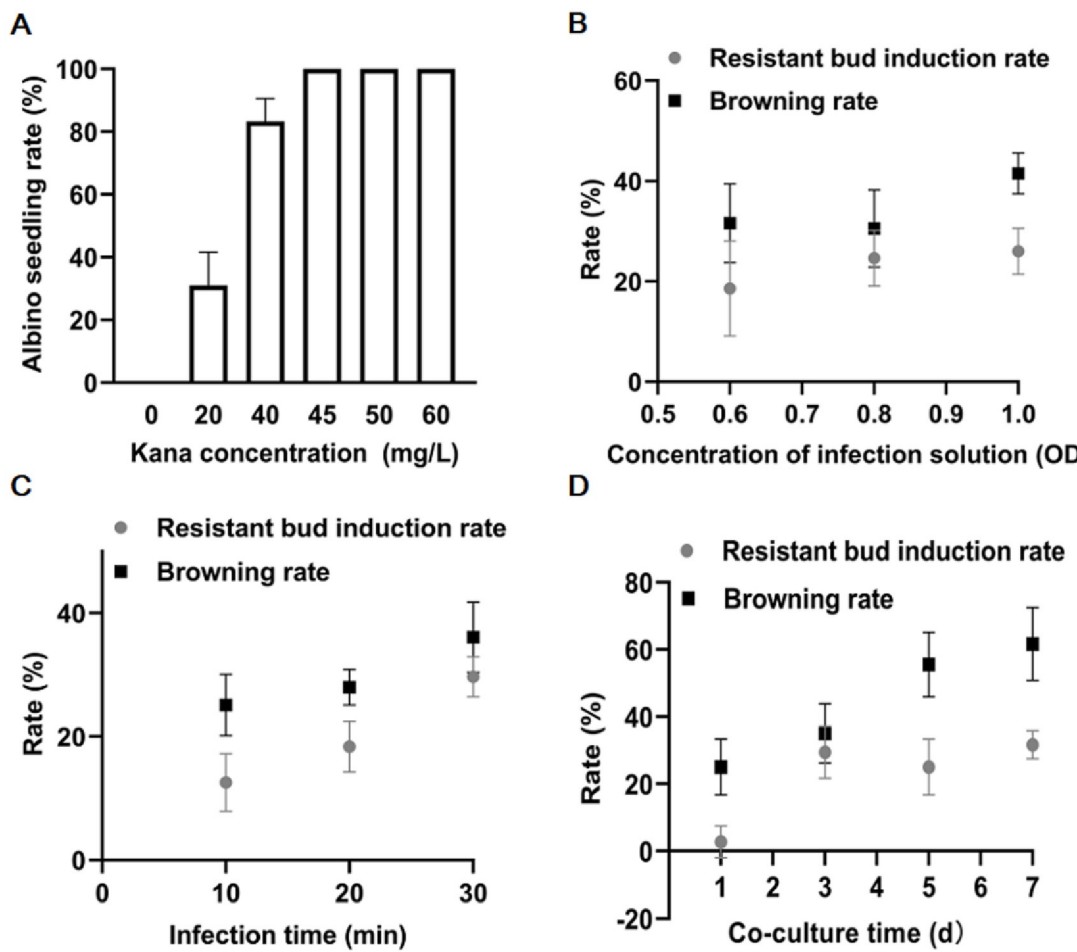

**Fig 5. The influent factors on transformation efficiency of ZADT.** (A) Determination of Kana concentration for screening resistant seedlings; (B-D) the effects of *Agrobacterium* density ($OD_{600}$), infection time and co-culture time on transformation efficiency of ZADT. The data were represented as the means±SE of three biological replicates.

establish and optimize this system for accelerating the breeding process of high yield, good quality and resistant varieties.

The selection of explants is an important step for the successful tissue regeneration for plants. In our study, direct induction of adventitious buds with cotyledonary nodes as explants was reached 94.4% with the highest efficiency, compared to these explants (hypocotyl, cotyledon, and true leaf). It has also been reported that cotyledonary node of soybean was selected to successfully induce regeneration [25]. Additionally, PGRs play an important role in inducing buds and roots with different explants. For example, callus tissues of duckweed (*Lemna aequinoctialis*) in Schenk and Hildebrandt (SH) medium containing 1 μmol/L 6-BA were differentiated into adventitious bud with 100% [26]; 2 mg/L 6-BA and 0.1 mg/L NAA in MS could induce buds with leaves of *Taraxacum officinale* [23]; 25 μmol/L thidiazuron (TDZ) and 10 μmol/L IAA in MS or B5 medium effectively induced adventitious buds with whole or half-leaf attached petiolar explants of *Paulownia elongata* [27]. The combination of IAA 0.05 mg/L and KT 2 mg/L efficiently induced cotyledonary nodes to differentiate adventitious in our study (Fig 1). At the stage of rooting, 0.9 mg/L IBA and 5 μmol/L IBA could induce the rooting of cotton [28] and *Paulownia elongata* [25] adventitious shoots, respectively. ZADT had the

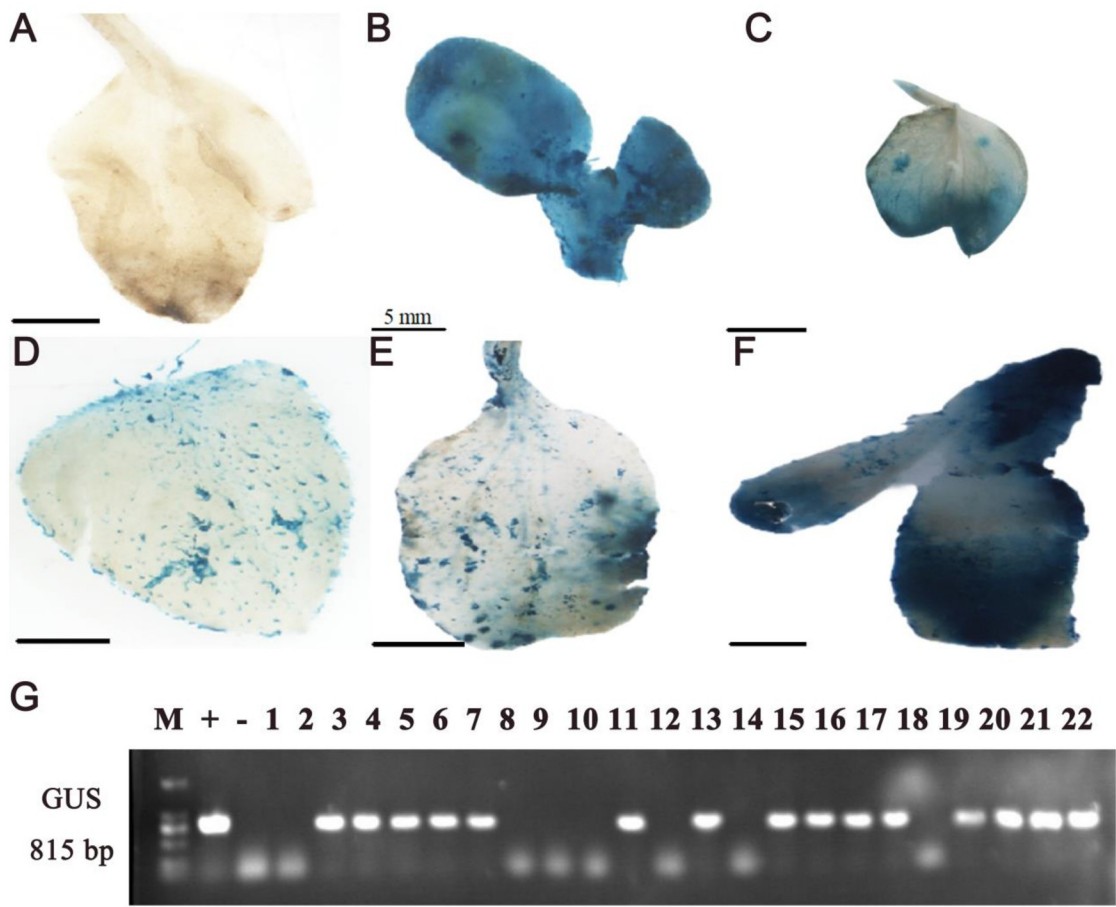

**Fig 6. GUS staining and *GUS* gene determination by genomic PCR with resistant seedlings of ZADT.** (A-F) GUS staining phenotype of control (A) and the leaves and buds of resistant seedlings (B-F). (G) the detection of *GUS* gene integration by genomic PCR. M: 2000 bp DNA marker, pBI121 plasmid (+), control (-), 1–22: tissues of resistant seedlings after kana screening.

highest rate of rooting, and the most developed root system was obtained with 0.9 mg/L IBA (Fig 2B).

About *Agrobacterium*-mediated genetic transformation for ZADT, three factors (*Agrobacterium* density, infection time, and co-culture time) were optimized by the number and phenotype of resistant buds. *Agrobacterium* density directly affected the amount of bacteria in contact with the explants; too many and too few bacteria were detrimental to plant transformation. Low inoculum levels and prolonged immersion of explants can also improve transformation efficiency in sunflower [29]. Additionally, the transfer of T-DNA by *Agrobacterium* to the plant genome is time-consuming; thus, the infection time and the co-culture time must affect transformation efficiency [30]. Differences in infection time are caused by variations in species and *Agrobacterium* strains. Therefore, the infection time of *Agrobacterium* strains needs to be optimized for different plants or even different genotypes of the same species [31, 32]. The co-culture time is also an important factor in influencing transformation efficiency [33–35]. In our study, the transmutation efficiency was the highest in 30 min of *Agrobacterium* strain *EHA105* in $OD_{600}$ 0.8, and co-culture was three days (Fig 4). In summary, the data indicated that we successfully established tissue culture and obtained an efficient genetic transformation system for ZADT.

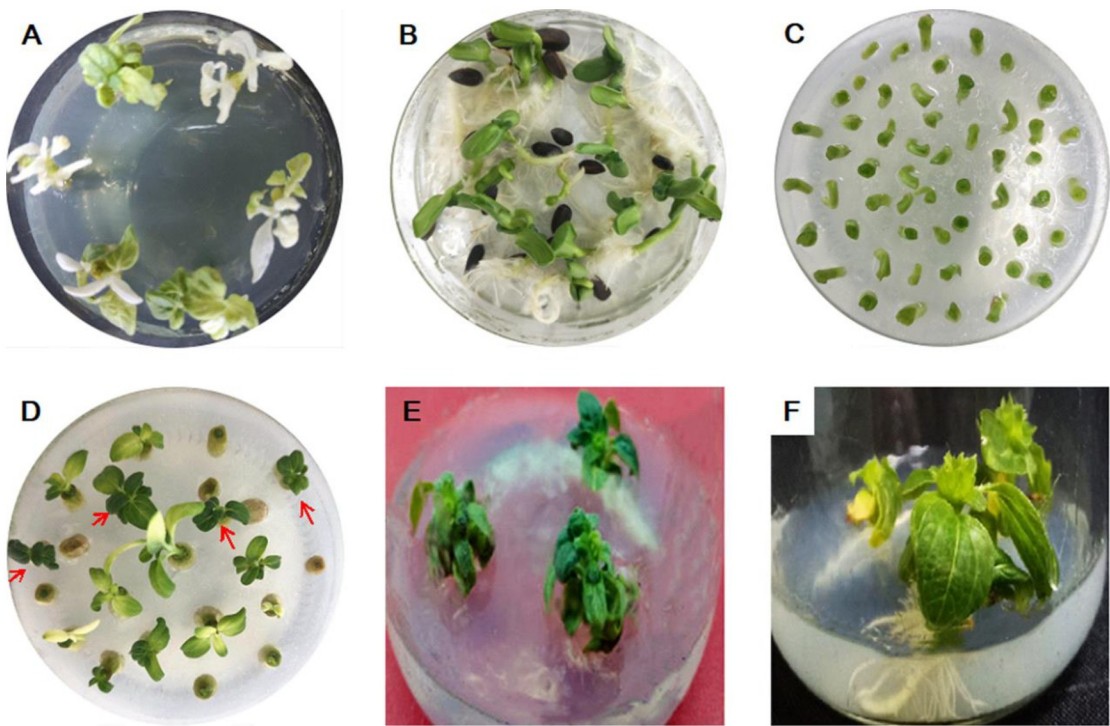

**Fig 7. The process of genetic transformation for ZADT.** (A) negative control; (B) 7-day-old sterile seedlings; (C) cotyledonary nodes were subjected to *Agrobacterium* infection for 30 min and transferred to co-culture medium for 3 days in the dark environments; (D) resistant buds were screened in MS medium containing kanamycine; (E) resistant seedlings were transferred to subculture medium for 2 weeks; (F) the rooting of resistant seedlings.

## Conclusion

This study established and optimized the regeneration and genetic transformation systems with the cotyledon nodes as explants from the retention line of backbone parent oil sunflower ZADT. The key steps in its tissue culture and regeneration were the hormone combination for the induction of adventitious buds was 0.05 mg/L IAA + 2 mg/L KT; the contents of IAA and KT hormones were halved for the subculture of adventitious buds and 0.9 mg/L IBA added in MS was used for rooting. For *Agrobacterium*-mediated genetic transformation for ZADT, the highest transformation efficiency was achieved at a bacterial density of $OD_{600}$ 0.8, infection time of 30 min, and co-cultivation of three days. The establishments of these two platforms for ZADT are very useful for the functions and mechnisms of genes and molecular breeding.

## Materials and methods

### Plant material preparation

Seeds of the retention line of backbone parent ZADT were obtained from the Institute of Economic Crops, Xinjiang Academy of Agricultural Sciences. Healthy seeds were surface sterilized with 75% alcohol for 1 min, commercial bleach (5% effective chlorine content) for 8 min, rinsed 5~6 times with sterile water and soaked for 8 h. Excess water on the surface of seeds was absorbed by sterilized filter paper and seeds were sown on 1/2 MS medium and incubated in a constant-temperature light incubator (light intensity: 2,000 lx, 28°C/16 h light, 26°C/8 h dark) for 7 d.

## Establishment and optimization of regeneration systems for ZADT

Some combinations of IAA and KT or IAA and 6-BA were designed (S1 and S2 Tables) to induce adventitious buds with these explants of cotyledons, hypocotyls, true leaves, and cotyledonary nodes incubated for 15 d. At least six replicates of each treatment with 6~7 explants per replicate were designed. The bud induction rate was calculated as follows:

the number of explants induced to generate buds (N2)/the number of explants (N1) $\times$ 100%.

The induced buds were cultured in the subculture medium (MS + 1/2 B5; pH 5.8) for 15 d and they were induced to root in these media (MS + 0.3 mg/L, 0.6 mg/L, 0.9 mg/L, 1.2 mg/L IBA, and pH 5.8) when buds reached the length of 1~3 cm. The rooting rate was calculated by formula: M2/M1×100% (M1: the total number of buds; M2: the number of buds induced rooting). And then, regenerated plantlet were domesticated outside, and grew in the substrate (perlite: vermiculite: flower soil = 1:1:3), normally.

## Determination of kanamycin concentration for screening resistant buds

The suitable kanamycin concentration among 0, 20, 40, 45, 50, and 60 mg/L was determined to screening resistant buds with cotyledonary nodes as explants. Cotyledonary nodes were put in the medium (MS + 0.05 mg/L IAA+ 2 mg/L KT+ 150 µmol/L As), incubated under natural light and temperature (26~28˚C) for 15 d. At least six replicates of each treatment with 6~7 explants per replicate were designed. The bleaching rate of the adventitious buds was counted.

## Structural composition of plant genetic transformation expression vectors and preparation of *Agrobacterium* immersion solution

*Agrobacterium tumefaciens* strain *EHA105* was used for the genetic transformation of ZADT. A binary vector pBI121 with a selection marker (a *GUS* reporter gene driven by the CaMV 35S promoter) and a *NPT II* (neomycin phosphotransferase II) resistance gene driven by the NOS (nopaline synthetase promoter) was introduced into *EHA105* by the freeze-thaw method.

For preparation of *Agrobacterium* immersion solution for ZADT, *Agrobacterium EHA105* carrying the pBI121 binary vector were inoculated in liquid YEB medium containing 50 mg/L kanamycin (Kana) and 25 mg/L rifampicin (rif) and grew at 28˚C for 16 h at a shaking frequency of 220 rpm. When the density of the bacterial solution was $OD_{600}$ 0.8, bacterial cells were collected through centrifugation at 5000 r/min for 15 min, and was resuspended three times with liquid MS medium containing 150 mg/L As [36].

## Evaluation of transformation efficiency by three factors

Transformation efficiency were evaluated by these three conditions of *Agrobacterium* density ($OD_{600}$ 0.6, 0.8, and 1.0), infection time (10, 20, and 30 min), and co-culture time (1, 3, 5, and 7 d). The detailed description was that the cotyledon nodes were gently shaken for 10, 20, or 30 min at 100 r/min in *Agrobacterium* immersion solution ($OD_{600}$ 0.6, 0.8, and 1.0), the surface was blotted with sterile filter paper and then cotyledon nodes were transferred to a co-culture medium (MS + 0.05 mg/L IAA + 2 mg/L KT + 150 µmol/L As + 7 g/L agar, and pH 5.8) for 1, 3, 5, and 7 d at 24˚C in the dark. The replicates was set for each treatment and 15 cotyledon nodes was for each replicate.

## Screening of the antibiotic plantlets

After the process of co-culture was finished, the cotyledonary nodes were cultured to generate resistant buds in a solid MS medium containing 500 mg/L Cefatothin Sodium (Cef) + 45 mg/L Kana + 0.05 mg/L IAA + 2 mg/L KT and were continuously subcultured in the medium (MS + 100 mg/L Cef + 20 mg/L Kana + 7 g/L agar, pH 5.8), and were rooted in the rooting medium (MS+100 mg/L Cef+0.9 mg/L IBA+7 g/L agar, pH 5.8) for two weeks each. Finally, they were domesticated and grew normally in the greenhouse.

## PCR assay and β-glucuronidase (GUS) staining

Genomic DNA from resistant buds of ZADT was isolated by the CTAB method and examined by PCR to confirm whether the *GUS* gene was integrated into the sunflower chromosomes. The specific primer sequences for amplifying the *GUS* gene were as follows: GUS-F: GCTATACGCCTTTGAAGCC and GUS-R: TTGACTGCCTCTTCGCTGTA.

The preliminary confirmation of the resistant buds of ZADT was positive by GUS staining, the leaves and buds of resistant buds were incubated in 100 mmol/L staining buffer overnight at 37˚C and then in 70% ethanol for 4 h to remove chlorophyll from the green tissue [37] and then these samples were observed and photographed.

## Data analysis

GraphPad Prism 8.0.2 and the SPSS statistics 20 were employed for data analysis. One-way analysis of variance (ANOVA) was use to evaluate the experimental data. All data were expressed as the means±standard error (SE) of the experiments, and Tukey's Least Significant Difference (LSD) test was performed to show significant differences by comparison. All differences were considered to be statistically significant at $p < 0.05$.

## Supporting information

**S1 Fig. The detection of GUS gene by genomic PCR analysis.**
(JPG)

**S2 Fig. Detection of the GUS gene of ZADT positive plants grown for 2 months.**
(JPG)

**S1 Table. Different hormone combinations induced adventitious shoots directly through three types of explants.**
(DOCX)

**S2 Table. Different hormone combinations induced adventitious shoots directly through cotyledonary nodes.**
(DOCX)

## Acknowledgments

We thank the Xinjiang Rapid Breeding Innovation Platform for the Molecular Design and Multilayer Genomic Linkage of Cash Crops, such as Cotton, Oilseed Rape, and Safflower, for providing vital instruments for conducting experiments. We are also grateful to Dr. Chengxia Lai and associate researcher Hong Sha for their helps in this study.

## Author Contributions

**Conceptualization:** Fangyuan Chen, Youling Zeng, Quan Cheng, Lvting Xiao, Jieyun Ji, Xianfei Hou, Qixiu Huang, Zhonghua Lei.

**Data curation:** Fangyuan Chen, Youling Zeng, Quan Cheng, Lvting Xiao, Jieyun Ji, Xianfei Hou, Qixiu Huang, Zhonghua Lei.

**Formal analysis:** Fangyuan Chen, Youling Zeng, Quan Cheng, Lvting Xiao, Jieyun Ji, Xianfei Hou, Qixiu Huang, Zhonghua Lei.

**Funding acquisition:** Fangyuan Chen, Youling Zeng.

**Writing – review & editing:** Fangyuan Chen, Youling Zeng.

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
