## [Decision Letter · Decision Letter 0]

16 Oct 2023

PONE-D-23-25867Tissue Culture and Agrobacterium-mediated Genetic Transformation of the Oil Crop SunflowerPLOS ONE

Dear Dr. chen,

Thank you for submitting your manuscript to PLOS ONE. After careful consideration, we feel that it has merit but does not fully meet PLOS ONE’s publication criteria as it currently stands. Therefore, we invite you to submit a revised version of the manuscript that addresses the points raised during the review process.

We look forward to receiving your revised manuscript.

Kind regards,

Vijay Kumar

Academic Editor

PLOS ONE

Journal Requirements:

 " This work was financially supported by the Natural Science Foundation of Xinjiang Uygur Autonomous Region (2020D01C020), Project 4 of Xinjiang Autonomous Region Major Special Project (2022A03004-4) and the Outstanding Doctoral Researchers Student Research Innovation Project in Xinjiang University (XJU2022BS050). They provided adequate funding for this study."

7. PLOS ONE now requires that authors provide the original uncropped and unadjusted images underlying all blot or gel results reported in a submission’s figures or Supporting Information files. This policy and the journal’s other requirements for blot/gel reporting and figure preparation are described in detail at https://journals.plos.org/plosone/s/figures#loc-blot-and-gel-reporting-requirements and https://journals.plos.org/plosone/s/figures#loc-preparing-figures-from-image-files. When you submit your revised manuscript, please ensure that your figures adhere fully to these guidelines and provide the original underlying images for all blot or gel data reported in your submission. See the following link for instructions on providing the original image data: https://journals.plos.org/plosone/s/figures#loc-original-images-for-blots-and-gels. 

Reviewers' comments:

Reviewer's Responses to Questions

**Comments to the Author**

1. Is the manuscript technically sound, and do the data support the conclusions?

Reviewer #1: Yes

Reviewer #2: Yes

Reviewer #3: No

2. Has the statistical analysis been performed appropriately and rigorously? 

Reviewer #1: Yes

Reviewer #2: Yes

Reviewer #3: No

3. Have the authors made all data underlying the findings in their manuscript fully available?

Reviewer #1: Yes

Reviewer #2: Yes

Reviewer #3: Yes

4. Is the manuscript presented in an intelligible fashion and written in standard English?

Reviewer #1: Yes

Reviewer #2: No

Reviewer #3: No

5. Review Comments to the Author

Reviewer #1: Gene transformation is a pivotal procedure for gene editing breeding. This paper use a dominate cultivar to establish the approach, which is usually a very challenging process. This work make the cultivar amenable to gene editing. This is very good.

There is no big problems in this paper. I would prefer an sound explanation why the blue color is evenly appeared on the whole tissue after GUS staining.

There also some minor writing formats errors. Please see below:

Some sentence are incomplete: "Next, were transferred to resistance screening“

Some braces lack space between the previous words: "medium(MS"

Some spaces also missed between two word.

Please make efforts on these writing errors.

Reviewer #2: The manuscript entitled “Tissue Culture and Agrobacterium-mediated Genetic Transformation of the Oil Crop Sunflower” studied regeneration and genetic transformation optimization in sunflower. The manuscript can be accepted after major revisions.

In the abstract section, “In this study, we selected different explants to establish a regeneration system and obtained that direct induction of adventitious buds and rooting by cotyledon nodes was in MS more suitable for the combination of 0.05 mg/L IAA 2.0 mg/L KT, and add 0.9 mg/L IBA.” This sentence needs to be rephrased for better understanding.

In the abstract section, “The rate of resistant seedlings as the highest transformation efficiency reached 8.0 % under the condition of 45 mg/L kanamycin, a bacterial density of OD600 0.8, infection time of 30 min and 3 days of co-cultivation” this sentence needs to change authors should mention optimization parameters in first then need to conclude the transformation efficiency.

The authors mentioned they used 150.0 mg/L Acetosyringone in transformation studies, which is very high; it should be in μM.

In Figure 4. Flow diagram of the tissue culture and regeneration for the ZADT. In the place of the Germination of sterile seeds image, the authors need to include the seedling image.

Figure 6. Expression of a GUS reporter gene from the 35S promoter in resistant seedlings of ZADT. GUS staining and PCR amplification images are poor; they must be replaced with good images.

Figure 8. Detection of the GUS gene of ZADT-positive plants grown for 2 months gel image is very poor, and a ladder needs to be added.

The author must clearly mention the selectable marker gene (nptII) name in the Genetic Transformation of Cotyledonary Nodes section.

Check the spelling of ‘convinient’ throughout the manuscript.

Check the spelling of ‘these system’ in the conclusion section.

Check the spelling of ‘flflower soil’ in the Regeneration optimization section.

The English language of the manuscript is very poor, and there are many grammatical mistakes that need to be corrected.

The authors calculated the transformation efficiency by GUS staining or PCR amplification.? please clarify.

Authors need to mention keywords in the manuscript, and it should be in alphabetical order.

Authors need to cite the latest articles instead of citing old articles.

Reviewer #3: Observation: Please split it into two parts for a better understanding of the sentence. “Plant regeneration protocol is usually a prerequisite to the genetic transformation,

the regeneration conditions are determined by optimising factors such as genotype,

explant type and concentration of the inducing hormones.”

Observation: Too lengthy sentence. Please split it. “Usually, genotype

specificity influences the regeneration of most plants, such as cotton[7], sunflower[6],

etc., so it is essential to develop a reliable and reproducible regeneration protocol for

different genotypes or varieties; mullein(Casuarina equisetifolia) stem segments can

induce of callus, adventitious buds and adventitious roots through different hormone

combinations[8], as do garland chrysanthemum[9].

Observation: Delete this part. It is not required. “plant growth regulators (PGRs) are synthetic chemicals that can replace its functions in vitro and are essential for the plant tissue culture process.”

Observation: Delete or rewrite this paragraph. “Among them, Agrobacterium-mediated method is the most practical and inexpensive technique for the most plants. The GUS encoding

β-glucuronidase[27] is often as a reporter gene to detect transformation efficiency by

histochemical staining and sensitive fluorescence quantification of enzyme activity.”

6. PLOS authors have the option to publish the peer review history of their article (what does this mean?). If published, this will include your full peer review and any attached files.

Reviewer #1: **Yes: **Fuqiang Cui

Reviewer #2: No

Reviewer #3: No

---

## [Author Response · Author response to Decision Letter 0]

5 Dec 2023

Dear academic editor Vijay Kumar ,

Thank you very much for your careful reading of this thesis and for your valuable comments. I have revised the content of the thesis in the light of the following issues.

I have reworked the article according to PLOS ONE's journal style formatting requirements.

I have re-checked and re-determined this section and made changes.

 " This work was financially supported by the Natural Science Foundation of Xinjiang Uygur Autonomous Region (2020D01C020), Project 4 of Xinjiang Autonomous Region Major Special Project (2022A03004-4) and the Outstanding Doctoral Researchers Student Research Innovation Project in Xinjiang University (XJU2022BS050). They provided adequate funding for this study."

This work was financially supported by the Natural Science Foundation of Xinjiang Uygur Autonomous Region (2020D01C020), Project 4 of Xinjiang Autonomous Region Major Special Project (2022A03004-4) and the Outstanding Doctoral Researchers Student Research Innovation Project in Xinjiang University (XJU2022BS050). The funders had no role in study design, data collection and analysis, decision to publish, or preparation of the manuscript.

The funder role statement changes have been added to the cover letter.

4.Your ethics statement should only appear in the Methods section of your manuscript. If your ethics statement is written in any section besides the Methods, please move it to the Methods section and delete it from any other section. Please ensure that your ethics statement is included in your manuscript, as the ethics statement entered into the online submission form will not be published alongside your manuscript.

My ethics statement has been moved to the methods section and removed from any other section.

5.Please include a separate caption for each figure in your manuscript.

I have added a separate caption for each figure in my manuscript.

6.Please include captions for your Supporting Information files at the end of your manuscript, and update any in-text citations to match accordingly. Please see our Supporting Information guidelines for more information: http://journals.plos.org/plosone/s/supporting-information.

I have added the support information at the end of the article. The additional data for this section is stored in the online site Figshare.

7.PLOS ONE now requires that authors provide the original uncropped and unadjusted images underlying all blot or gel results reported in a submission’s figures or Supporting Information files. This policy and the journal’s other requirements for blot/gel reporting and figure preparation are described in detail at https://journals.plos.org/plosone/s/figures#loc-blot-and-gel-reporting-requirements and https://journals.plos.org/plosone/s/figures#loc-preparing-figures-from-image-files. When you submit your revised manuscript, please ensure that your figures adhere fully to these guidelines and provide the original underlying images for all blot or gel data reported in your submission. See the following link for instructions on providing the original image data: https://journals.plos.org/plosone/s/figures#loc-original-images-for-blots-and-gels. 

 I have submitted the original uncropped and unadjusted images of the gel to the online site figshare as required by the journal and mentioned in the cocverletter.

5. Review Comments to the Author

Reviewer #1: Gene transformation is a pivotal procedure for gene editing breeding. This paper use a dominate cultivar to establish the approach, which is usually a very challenging process. This work make the cultivar amenable to gene editing. This is very good.

There is no big problems in this paper. I would prefer an sound explanation why the blue color is evenly appeared on the whole tissue after GUS staining.

Thank you for your question. This is because the vector pBI121 used on our transgene contains a 35S strong constitutive promoter. Changing the promoter enables the GUS gene to initiate expression in all tissues.

There also some minor writing formats errors. Please see below:

Some sentence are incomplete: "Next, were transferred to resistance screening“

Some braces lack space between the previous words: "medium(MS"

Some spaces also missed between two word.

Please make efforts on these writing errors.

These errors were created due to our negligence and we have corrected them accordingly.

Reviewer #2: The manuscript entitled “Tissue Culture and Agrobacterium-mediated Genetic Transformation of the Oil Crop Sunflower” studied regeneration and genetic transformation optimization in sunflower. The manuscript can be accepted after major revisions.

In the abstract section, “In this study, we selected different explants to establish a regeneration system and obtained that direct induction of adventitious buds and rooting by cotyledon nodes was in MS more suitable for the combination of 0.05 mg/L IAA 2.0 mg/L KT, and add 0.9 mg/L IBA.” This sentence needs to be rephrased for better understanding.

In the abstract section, “The rate of resistant seedlings as the highest transformation efficiency reached 8.0 % under the condition of 45 mg/L kanamycin, a bacterial density of OD600 0.8, infection time of 30 min and 3 days of co-cultivation” this sentence needs to change authors should mention optimization parameters in first then need to conclude the transformation efficiency.

 Thank you for your detailed questions that have made this article better. We have revised them one by one.

The authors mentioned they used 150.0 mg/L Acetosyringone in transformation studies, which is very high; it should be in μM.

 These errors were created due to our negligence and we have corrected them accordingly.

In Figure 4. Flow diagram of the tissue culture and regeneration for the ZADT. In the place of the Germination of sterile seeds image, the authors need to include the seedling image.

This image has been modified as per the suggestions you have given.

Figure 6. Expression of a GUS reporter gene from the 35S promoter in resistant seedlings of ZADT. GUS staining and PCR amplification images are poor; they must be replaced with good images.

We were fortunate to find genomes of plant material extracted from previous experiments. The images have been re-replaced.

Figure 8. Detection of the GUS gene of ZADT-positive plants grown for 2 months gel image is very poor, and a ladder needs to be added.

We were fortunate to find genomes of plant material extracted from previous experiments. The images have been re-replaced.

The author must clearly mention the selectable marker gene (nptII) name in the Genetic Transformation of Cotyledonary Nodes section.

 Based on your suggestions, we have made changes in the corresponding places.

Check the spelling of ‘convinient’ throughout the manuscript.

Check the spelling of ‘these system’ in the conclusion section.

Check the spelling of ‘flflower soil’ in the Regeneration optimization section.

The English language of the manuscript is very poor, and there are many grammatical mistakes that need to be corrected.

Thanks to your suggestion, we have re-corrected all of this article for word and sentence grammar.

The authors calculated the transformation efficiency by GUS staining or PCR amplification.? please clarify.

We calculated the transformation efficiency by PCR amplification. This part we have revised and mentioned at the right place in the article.

Authors need to mention keywords in the manuscript, and it should be in alphabetical order.

Thanks for the reminder that the keywords have been added at the bottom of the abstract.

Authors need to cite the latest articles instead of citing old articles.

Regarding citations, we have attention to new and relevant literature reports. It is estimated that it is still incomplete, and we have added the latest reported literature in this revision. However, regarding sunflower-related reports, there are fewer recent studies, mainly some earlier reports.

Reviewer #3: Observation: Please split it into two parts for a better understanding of the sentence. “Plant regeneration protocol is usually a prerequisite to the genetic transformation, the regeneration conditions are determined by optimising factors such as genotype, explant type and concentration of the inducing hormones.”

Observation: Too lengthy sentence. Please split it. “Usually, genotype

---

## [Decision Letter · Decision Letter 1]

24 Mar 2024

PONE-D-23-25867R1Tissue culture and *Agrobacterium*-mediated genetic transformation of the oil crop sunflowerPLOS ONE

Dear Dr. Chen,

Thank you for submitting your manuscript to PLOS ONE. After careful consideration, we feel that it has merit but does not fully meet PLOS ONE’s publication criteria as it currently stands. Therefore, we invite you to submit a revised version of the manuscript that addresses the points raised during the review process.

We look forward to receiving your revised manuscript.

Kind regards,

Jianhong Zhou

Staff Editor

PLOS ONE

Journal Requirements:

**Additional Editor Comments:**

Specifically, please address reviewer 1's comments about the uneven staining.

Reviewers' comments:

Reviewer's Responses to Questions

**Comments to the Author**

1. If the authors have adequately addressed your comments raised in a previous round of review and you feel that this manuscript is now acceptable for publication, you may indicate that here to bypass the “Comments to the Author” section, enter your conflict of interest statement in the “Confidential to Editor” section, and submit your "Accept" recommendation.

Reviewer #1: All comments have been addressed

Reviewer #2: All comments have been addressed

2. Is the manuscript technically sound, and do the data support the conclusions?

Reviewer #1: Partly

Reviewer #2: Yes

3. Has the statistical analysis been performed appropriately and rigorously? 

Reviewer #1: Yes

Reviewer #2: Yes

4. Have the authors made all data underlying the findings in their manuscript fully available?

Reviewer #1: Yes

Reviewer #2: Yes

5. Is the manuscript presented in an intelligible fashion and written in standard English?

Reviewer #1: Yes

Reviewer #2: Yes

6. Review Comments to the Author

Reviewer #1: Dear Chen et al.,

I am sorry for my typos, while I have to ask you again for the question: why the

blue color is NOT evenly appeared on the whole tissue after GUS staining.

Otherwise, it's a good paper and revision.

Reviewer #2: (No Response)

7. PLOS authors have the option to publish the peer review history of their article (what does this mean?). If published, this will include your full peer review and any attached files.

Reviewer #1: No

Reviewer #2: No

---

## [Author Response · Author response to Decision Letter 1]

1 Apr 2024

Dear Dr Zhou Jianhong, Editorial Board Member, PLoS One and Peer reviewers： 

I am youling Zeng, as a correponding author for this manuscript. We greatly appreciate the comments from you and the reviewers on our paper. We have revised our manuscript carefully as possibly again.

Enclosed please find our revised manuscript (ID:PONE-D-23-25867R1), entitled “Tissue culture and Agrobacterium-mediated genetic transformation of the oil crop sunflower”. In the revised manuscript, we have addressed the comments from you and the reviewer. We are looking forward to hearing good news on publishing this paper in the PLoS One journal.

Sincerely yours,

Corresponding author: Dr. Zeng Youling

e-mail: zeng_ylxju@126.com

Xinjiang Key Laboratory of Biological Resources and Genetic Engineering, College of Life Science and Technology, Xinjiang University, 777 Huarui Street, Urumqi 830091, China

On the comments from Reviewer:

 Thanks. We have checked the cited literature one by one. The main modifications are the invalid citation deletion, symbols standardization, page number formatting modification and article information checking, and currently, our manuscript has met the requirements of this journal by our proofreads. These modifications can be viewed in the version of the modification mode.

 Invalid literature removed included:

2. Zhang HM, ZhaoY, Zhu JK. Thriving under stress: how plants balance growth and the stress response. Cell. 2020; 55:529–43. https://doi.org/10.1016/j.devcel.2020.10.012 PMID: 33290694 

3. Liang M, Mi XJ, Li CH, Zhao J, Wang YG, Ma J, et al. Salinity characteristics and halophytic vegetation diversity of uncultivated saline-alkali soil in Junggar Basin, Xinjiang. Arid Land Geography. 2021; 45(1):185–96.

5. Mohamed S, Boehm R, Schnabl H. Stable genetic transformation of high oleic Helianthus annuus L. genotypes with high efficiency. Plant Sci, Oxford. 2006; 171(5):546–54.

21. Wang ZK, Wang YZ, Shang P, Yang C, Yang MM, Huang JX, et al. Overexpression of soybean GmWRI1a stably increases the seed oil content in soybean. Int. J. Mol. Sci. 2022; 23(9):5084–100. https://doi.org/10.3390/ijms23095084 PMID:35563472

23. Wang M, Sun RR, Zhang BH, Wang QL. Pollen tube pathway-mediated cotton transformation. Methods Mol Biol. 2019; 1902:67–73. https://doi.org/10.1007/978–1–4939–8952–2_6. PMID: 30543062

32. Schrammeijer B, Sijmons PC, Elzen PJM, Hoekema A. Meristem transforrnation of sunflower via Agrobacterium. Plant Cell Rep. 1990; 9(2):55–60. https://doi.org/10.1007/BF00231548 PMID: 24226429

41. Jefferson RA, Kavanagh TA, Bevan MW. GUS fusions: β-glucuronidase as a sensitive and versatile gene fusion marker in higher plants. Embo J. 1987; 6(13):3901–7. https://doi.org/10.1002/j.1460–2075.1987.tb02730.x PMID: 3327686

45. He T, Vaidya BN, Perry ZD, Parajuli P, Joshee N. Paulownia as a medicinal tree: traditional uses and current advances. Eur. J. Med. Plants. 2016, 14(1):1–15.

48. Manickavasagam M, Subramanyam K, Ishwarya R, Elayaraja D, Ganapathi A. Assessment of factors influencing the tissue culture-independent Agrobacterium-mediated in planta genetic transformation of okra [Abelmoschus esculentus (L.) Moench]. Plant Cell, Tiss Org, 2015; 123(2):309–20.

 54. Nyaboga E, Tripathi JN, Manoharan R, Tripathi L. Agrobacterium-mediated genetic transformation of yam (Dioscorea rotundata): an important tool for functional study of genes and crop improvement. Front Plant Sci, 2014; 5:463. https://doi.org/10.3389/fpls.2014.00463 PMID: 25309562

55. Yang XF, Yu XQ, Zhou Z, Ma WJ, Tang GX. A high-efficiency Agrobacterium tumefaciens mediated transformation system using cotyledonary node as explants in soybean (Glycine max L.). Acta physiol plant, 2016; 38(3):1–10.

56. Rahman ZA, Seman ZA, Basirun N, Julkifle AL, Zainal Z, Subramaniam S. Preliminary investigations of Agrobacterium-mediated transformation in indica rice MR219 embryogenic callus using gusA gene. Afr J Biotechnol, 2011; 10(40):7805–13.

In addition, new additions to the literature include:

5.Sundararajan S, Sivaraman B, Rajendran V, Ramalingam S. Tissue culture and Agrobacterium-mediated genetic transformation studies in four commercially important indica rice cultivars. J Crop Sci Biotech. 2017; 20(3):175–83. https://doi.org/10.1007/s12892-017-0045-0

31.Gelvin S B. Integration of Agrobacterium T-DNA into the plant genome. Annu Rev Genet, 2017; 51:195–217. https://doi.org/10.1146/annurev-genet-120215-035320

38. Jefferson RA, Kavanagh TA, Bevan MW. GUS fusions: beta Glucuronidase as a sensitive and versatile gene fusion marker in higher plants. EMBO J. 1987; 6:3901–7. https://doi: 10.1002/j.1460-2075.1987.tb02730.x

On the comments from Editor:

Specifically, please address reviewer 1's comments about the uneven staining.

Reviewer #1: Dear Chen et al.,

I am sorry for my typos, while I have to ask you again for the question: why the

blue color is NOT evenly appeared on the whole tissue after GUS staining.

Otherwise, it's a good paper and revision.

Thank you very much for your question. In this experiment, although the constitutive promoter CaMV 35S was used to control the express of the GUS gene, we observed the uneven staining in transformed sunflower tissues indeed. In some literatures[1, 2], the same appearance for GUS staining was also observed. We can speculate inconsistency staining of different parts maybe due to the fact that the metabolic environments of the cells in different parts are not the same, which leads to certain differences in gene expression and then the staining of different sites was inconsistent.

Reference

[1] Udayabhanu J, Huang TD, Xin SC, Cheng J, Hua YW, Huang HS. Optimization of the transformation protocol for increased efficiency of genetic transformation in Hevea brasiliensis. Plants, 2022, 11, 1067. https://doi.org/10.3390/ plants11081067

[2] Ganguly S, Ghosh G, Ghosh S, Purohit A, Chaudhuri RK, Das S, Chakraborti D. Plumular meristem transformation system for chickpea: an efficient method to overcome recalcitrant tissue culture responses. PCTOC, 2020, 142:493-504. https://doi.org/10.1007/s11240-020-01873-8

---

## [Editor Report · Decision Letter 2]

11 Apr 2024

Tissue culture and *Agrobacterium*-mediated genetic transformation of the oil crop sunflower

PONE-D-23-25867R2

Dear Dr. Chen,

We’re pleased to inform you that your manuscript has been judged scientifically suitable for publication and will be formally accepted for publication once it meets all outstanding technical requirements.

Kind regards,

Jianhong Zhou

Staff Editor

PLOS ONE